# Interval timing in deep reinforcement learning agents

**Ben Deverett**
DeepMind
bendeverett@google.com

**Ryan Faulkner**
DeepMind
rfaulk@google.com

**Meire Fortunato**
DeepMind
meirefortunato@google.com

**Greg Wayne**
DeepMind
gregwayne@google.com

**Joel Z. Leibo**
DeepMind
jzl@google.com

## Abstract

The measurement of time is central to intelligent behavior. We know that both animals and artificial agents can successfully use temporal dependencies to select actions. In artificial agents, little work has directly addressed (1) which architectural components are necessary for successful development of this ability, (2) how this timing ability comes to be represented in the units and actions of the agent, and (3) whether the resulting behavior of the system converges on solutions similar to those of biology. Here we studied interval timing abilities in deep reinforcement learning agents trained end-to-end on an interval reproduction paradigm inspired by experimental literature on mechanisms of timing. We characterize the strategies developed by recurrent and feedforward agents, which both succeed at temporal reproduction using distinct mechanisms, some of which bear specific and intriguing similarities to biological systems. These findings advance our understanding of how agents come to represent time, and they highlight the value of experimentally inspired approaches to characterizing agent abilities.

## 1 Introduction

To exploit the rewards available in our environment, we capitalize on relationships between environmental causes and effects that exhibit precise temporal dependencies. For example, to avoid a dangerous threat moving towards you, you may estimate its speed by observing its displacement over a fixed time interval, extrapolate its future position over another time interval, and condition your escape behavior on your estimated time of contact with the threat. This ability to measure time and use it to guide behavior is necessary and prevalent in both animals and artificial agents. However, owing to basic differences in their implementation, artificial intelligence (AI) and biology have different relationships to time. Nevertheless, it is likely that consideration of the temporal measurement problem across these domains may yield valuable insights for both.

In biological systems, time measurements are necessary at a variety of temporal scales, ranging from milliseconds to years. The mechanisms underlying these timing abilities differ according to time scale, and many are well characterized at the level of the neural circuits [8, 24]. On the scale of seconds, interval timing paradigms are used in animals to study the behavioral and neural properties of time measurement. For example, an animal might be taught to measure out the elapsed interval between two events, then to report or reproduce that interval to the best of their ability in order to obtain a reward [3]. Humans, non-human primates, and rodents exhibit a number of characteristic behaviors on these tasks [5, 14, 13] that may reflect biological constraints on mechanisms that remain incompletely understood.

In the AI domain, there exist numerous agents that have succeeded in solving tasks with complex temporal dependencies [31, 29, 30, 11, 12]. Many of these are in the category of deep reinforcement learning agents, which develop reinforcement learning policies that use deep neural networks as function approximators [21]. While the abilities of these agents have advanced dramatically in recent years, we lack detailed understanding of the solutions they employ.

For instance, consider an agent that must learn to condition its actions on the amount of elapsed time between two environmental stimuli, as we will do in this study. A deep reinforcement learning agent with a recurrent module (e.g. LSTM [10]) has, by construction, two distinct mechanisms for storing relevant timing information. First, the LSTM is a source of temporal memory, since it is designed to store past information in model parameters trained by way of backpropagation through time. Second, the reinforcement learning algorithm, regardless of the underlying function approximator, assigns the credit associated with rewards to specific past states and actions. A deep reinforcement learning agent without a recurrent module (i.e. purely feedforward) lacks the former mechanism but retains the latter one. When trained end-to-end on a timing task, it is unclear whether and how agents may come to implicitly or explicitly represent time.

Here we use an experimentally inspired approach to study the solutions that reinforcement learning agents develop for interval timing. We characterize how the strategies developed by the agents differ from one another, and from animals, discovering themes that underlie interval timing. We suggest that this approach offers benefits both for AI – by introducing an experimental paradigm that simply and precisely evaluates agent strategies, and for neuroscience – by exploring the space of solutions that develop outside of biological constraints, serving as a testbed for interpretation of timing-related findings in animals.

## 2  Methods

### 2.1  Interval reproduction task

We designed a task based on a temporal reproduction behavioral paradigm in the neuroscience literature [14]. The task was implemented in PsychLab [19], a simulated laboratory-like environment inside DeepMind lab [2] in which agents view a screen and make "eye" movements to obtain rewards. We have open-sourced the task (along with other related timing tasks) for use in future work.

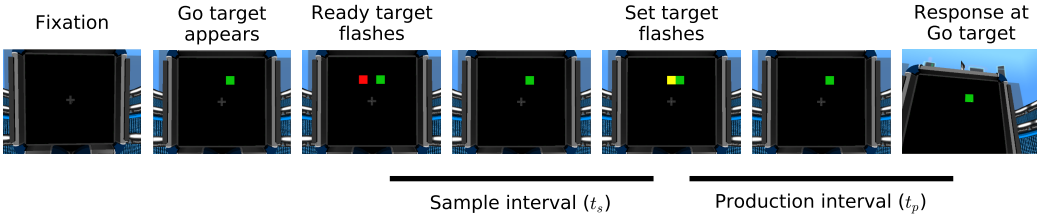

Figure 1: Interval reproduction task. The image sequence shows a single trial of the task. First, the agent fixates on a center cross, at which point the "Go" target appears. After a short delay, the red "Ready" cue flashes, followed by a randomly chosen "sample interval" delay. After the sample interval passes, the yellow "Set" cue flashes. Then the agent must wait for duration of the sample interval before gazing onto the "Go" target to end the trial. If the period over which it waited, the "production interval", matches the sample interval within a tolerance, the agent is rewarded. This task is closely based on an existing temporal reproduction task for humans and non-human primates [14].

The task is shown in Fig. 1. In each trial, the agent fixates on a central start position, at which point a "Go" target appears on the screen, which will serve as the eventual gaze destination to end the trial. After a delay, a "Ready" cue flashes, followed by a specific "sample" interval of time, then a "Set" cue flashes. Following the flash of the "Set" cue, the agent must wait for the duration of the sample interval before gazing onto the "Go" target to complete the trial. If the duration of the "production" interval (i.e. the elapsed time from "Set" cue appearance until gaze arrives on "Go" target) matches the sample interval within a specified tolerance, the agent is rewarded.

The demands of this "temporal reproduction" task are twofold: the agent must first measure the temporal interval presented between two transient environmental events, and it then must reproduce that interval again before ending the trial. Trials are presented in episodes, with each episode containing 300 seconds or a maximum of 50 trials, whichever comes first. Each trial's sample interval is selected randomly from the uniform range from 10-100 frames in steps of 10 (corresponding to 167-1667 ms at 60 frames per second). The agent is rewarded if the production interval is sufficiently close to the sample interval; specifically, if $|t_p - t_s| < \gamma_s(\alpha + \beta t_s)$, where $t_p$ is the production interval, $t_s$ is the sample interval, $\alpha$ is a baseline tolerance, $\beta$ is a scaling factor like that used in [14] to account for scalar variability, and $\gamma_s$ is an overall difficulty scaling factor for each sample interval $s$. In practice, we usually set $\alpha$ to 8 frames, $\beta$ to zero, and $\gamma_s$ evolved within an episode from 2.5 to 1.5 to 0, advancing each time two rewards were obtained at the given sample interval $s$. In practice we found that the results shown are robust to a wide range of parameters for this curriculum.

## 2.2 Agent architecture

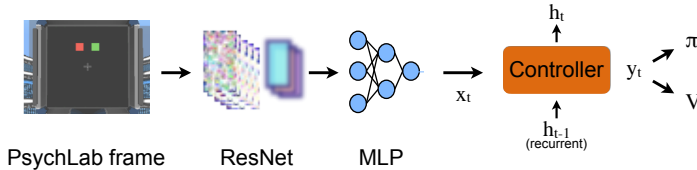

Figure 2: The agent architecture for our interval timing tasks. Frame input is passed into the residual network + MLP. The output from this component is passed to the controller, which enables the integration of past events in the recurrent case. Finally, this output is sent to the policy and value networks to generate an action (and a policy gradient in the backward pass).

We used an agent based on the A3C architecture [21] (Fig. 2). It uses a deep residual network [9] to generate a latent representation of the visual input from Psychlab which is subsequently passed to a controller network: either a recurrent network, in this case an LSTM, or a feed-forward network. The controller output is then fed forward to the policy and baseline networks that generate policy and value estimates that are then trained under the Importance Weighted Actor-Learner Architecture [6] (see section A.1). At each time step, the policy generates an action, corresponding to a small instantaneous eye movement in a particular direction from its current position. The LSTM controller provides a way for the agent to integrate past events along with its input in order to drive the policy while the feedforward agent must rely explicitly on the state of the environment to select actions.

We chose a residual embedding network architecture composed of three convolutional blocks with feature map counts of 16, 32, and 32; each block has a convolutional layer with kernel size 3x3 followed by max pooling with kernel size 3x3 and stride 2x2, followed by two residual subblocks. The ResNet was followed by a 256-unit MLP. We used controllers with 128 hidden units for all experiments. The learner was given trajectories of 100 frames, with a batch size of 32, and used 200 actors. Other parameters were a discount factor of 0.99, baseline cost of 0.5, and entropy cost of 0.01. The model was optimized using Adam with $\beta_1 = 0.9$, $\beta_2 = 0.999$, $\epsilon = 10^{-4}$, and a learning rate of $10^{-5}$.

## 3 Results

### 3.1 Performance of deep reinforcement learning agents

The recurrent agent learned to perform the task with near-perfect accuracy (Fig. 3a, b; top row); in other words, the production interval was matched to the sample interval across all presented sample intervals. From this analysis, however, it remains unclear whether the agent learned a general timing rule, or whether it memorized a specific discrete set of durations. Fig. 3c demonstrates that the agent indeed learned a general rule, successfully interpolating and extrapolating to new sample intervals on which it was not trained (+ signs in Fig. 3c).

Somewhat surprisingly, the feedforward agent also learned to performed the task, albeit more slowly and to a lesser degree of accuracy (Fig. 3, bottom row). The feedforward agent exhibited some

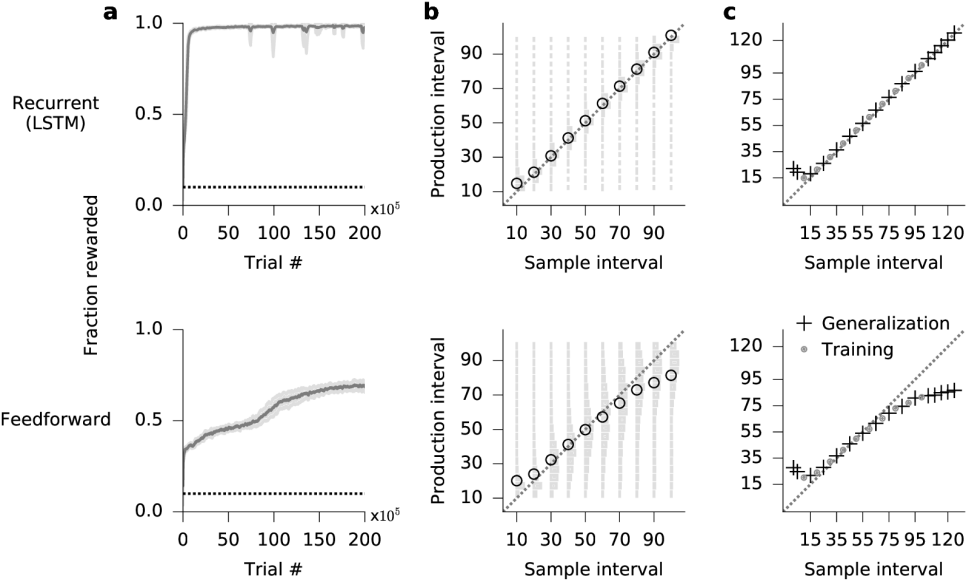

Figure 3: Agent performance on interval reproduction task. (a) Reward rate over training for the recurrent and feedforward agents. Lines show mean ± s.d. over 15 seeds. (b) Mean production interval in the trained agent for each of the ten unique sample intervals. Underlying gray histograms show the distribution of production intervals. Includes data from one actor in the final 60,000 trials only, thus excluding the initial training phase. (c) Generalization was assessed by presenting sample intervals not used to train the agent (+ signs).

notable behavioral features relative to the recurrent agent: (1) production interval distributions were wider, (2) a mean-directed bias was found on the production intervals at the extremes of the sample interval distribution, and (3) generalization to untrained intervals was poorer. Nevertheless, the agent learned to produce intervals that were remarkably well matched to the sample intervals, especially given the absence of any traditional or explicit memory systems within the agent architecture.

## 3.2 Psychophysical model of feedforward agent

One notable feature of the feedforward agent's solution was its similarity to human and primate data [14, 13]. In particular, the sigmoid-like shape of the performance curve suggests that the strategy might be well explained by established models of perceptual timing in humans. We tested this intuition quantitatively, since an alignment between agent and animal performance could draw useful links between analyses of agent and animal behaviors.

We fit the feedforward agent data to a Bayesian psychophysical model previously established for human [13] and non-human primate [14] studies. In brief, the model treats the task as a three-stage process: a noisy observation of a sample interval $t_s$ measured as $t_m$, a Bayesian least squares estimation $t_e$ of the true $t_s$ given the noisy measurement $t_m$, then the generation of a noisy production interval $t_p$ from the estimated interval $t_e$. The measurement and production steps are modeled as Gaussians with one parameter each, $w_m$ and $w_p$ respectively, corresponding to the coefficient of variation controlling the scalar variability [7] in the noisy measurement and production processes. The conditional probability of a given production interval $p(t_p|t_s, w_m, w_p)$ can then be computed by marginalizing over the intermediate distributions as described in the full model, found in [13]. The model was fit using optimization routines in Scipy [15].

Fig. 4a shows that in the feedforward agent, the standard deviation of the production interval scales with the sample interval. While this relationship is slightly sub-linear, it approximates scalar variability, a feature used to motivate the psychophysical model because of its prevalence in biological systems [7]. Fig. 4b shows the model fit and the agent data; the approximate alignment of these data, and in particular the mean-directed bias at the tails, suggests an alignment between agent and animal behaviors. While such models are known have poor identifiability [1], the qualitative alignment of

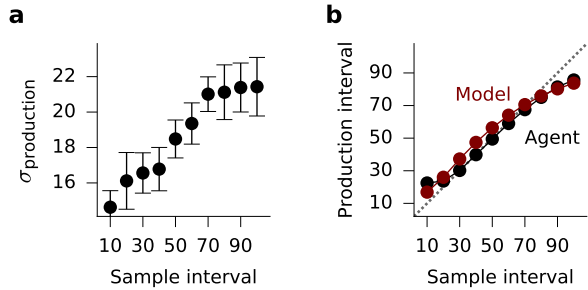

Figure 4: Psychophysical model of interval timing. (a) In the trained agent, the standard deviation of the production interval scales with interval duration. When fit to the power law $y = a + bx^c$, the best-fit value for $c$ is 0.7. Error bars: s.e.m. over 3 seeds. (b) An established Bayesian psychophysical model was fit to the feedforward agent data.

these results indicates the relevance of these animal-model tools for characterizing artificial agent behavior as well.

### 3.3  Evaluation of hidden unit activations

To understand the mechanism used by an agent to solve the task, it is helpful to characterize the activity of the hidden units [31]. In this task, it is reasonable to predict that the agent's hidden units should encode the timing information that the agent has learned in a trial. In particular, in the recurrent agent, one might predict the development of a "timer", in which the activations of one or many neurons implement a counter that accumulates over the presentation of the sample interval then reads out its value to produce the interval of interest.

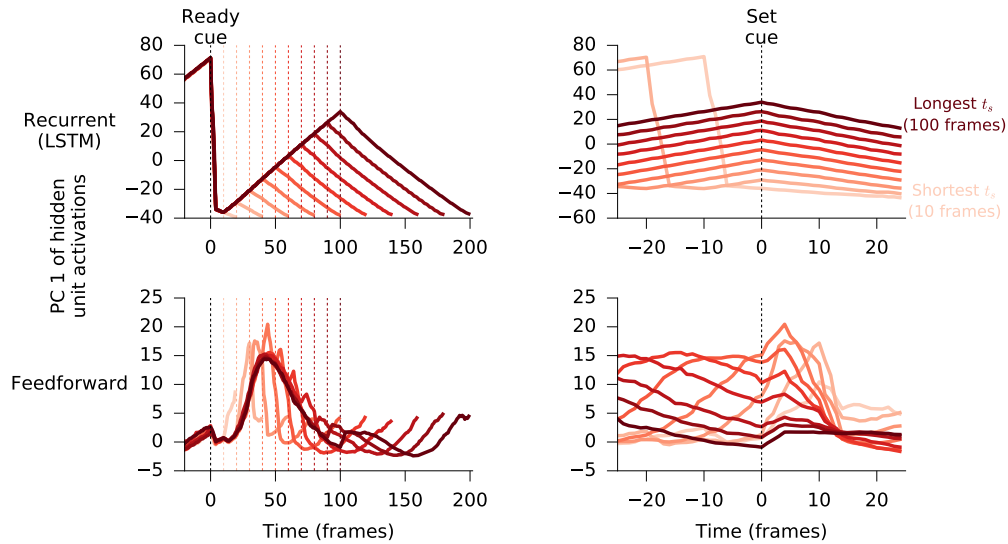

Figure 5: Hidden unit representations of time. The mean activations of the 128 hidden units (top row: LSTM cell state in recurrent agent; bottom row: hidden units in feedforward agent) are shown for trials of each sample interval duration. The activations of the population of hidden units are summarized by the first prinicpal component. Each color corresponds to trials of one specific sample interval duration (darker colors correspond to longer sample intervals). Left column: activations temporally aligned to the Ready cue; colored dashed lines: onset of each respective Set cue. Right column: same data aligned to Set cue.

In Fig. 5 (top row) we show that indeed such counters can be found in the unit activity of the trained recurrent agent. We summarize the unit activity of the 128 LSTM cell state units using their first principal component. During presentation of the sample interval, unit activity rises uniformly, until

the *Set* cue is presented, at which point activity begins to fall, reaching its initial value by the time that duration passes again. This is a simple solution to encoding the time interval and represents a form of clock. As a consequence, it can be seen that the Set-cue-aligned activity separates trials of different duration (Fig. 5, upper right), such that activity falls from a higher set point when the target interval is longer. These aligned average traces bear close resemblance to the unit activity found empirically in non-human primate parietal cortex during interval timing [14].

In feedforward agents, however, it is less clear how the activations of the neurons might represent the timing information. Using the same analysis on the hidden units of the feedforward agent, we found that unit activity also represents intervals, however in a less straightforward way (Fig. 5, bottom). To gain a better understanding of the feedforward solution, we proceeded to analyze the actions of the agent.

## 3.4 Action trajectories

The success of the feedforward agent is intriguing because it suggests the agent has learned a strategy that requires no persistent internal information, but rather achieves clock-like functionality using only its trained feedforward weights and external input. In order to characterize the strategies the agents developed and how they differed from one another, we evaluated the trajectories of the agents in action space.

The use of highly controlled stimuli and actions in our task, inspired by neuroscience literature, allows us to perform this analysis in a straightforward way. Because our stimuli were presented on a 2D screen and the only allowed actions were shifts in gaze direction, we could simply analyze the agent's gaze aligned to moments of interest in trials of each sample interval duration. In Fig. 6 we show this analysis. In the recurrent agent, action trajectories appear similar across the range of sample intervals: the agent maintains fixation in a small region near the initial fixation point throughout the Ready-Set interval (which it must measure), then it linearly shifts gaze to align with the target at the desired time.

On the other hand, the feedforward agent shows a more interesting pattern: after the Ready cue, it begins to traverse a stereotyped trajectory. When the Set cue arrives, it deviates off the trajectory and proceeds along another stereotyped trajectory, which it follows until it reaches the target. By expanding the extent of these trajectories in a consistent way, the agent measures elapsed time. This strategy can be described as one that uses the external environment as a clock, which is rational in the absence of any persistent internal states to use as a clock.

One framework that may explain this feedforward agent's solution has been studied in animal behavior research and is called stigmergy [28]: coordination with the external environment to indirectly transfer information across individuals. In this case, one may describe the stereotyped action pattern used for interval timing as "autostigmergy": the agent's own interactions with the environment serve as a source of memory external to the agent, but which can nevertheless be used to guide its actions. This "autostigmergic" solution is a particularly interesting proposition in light of the existing literature on mechanisms of timing in animals: many studies have suggested that animals may measure and encode time through a process inherently linked with their behavior [18]. In other words, rather than explicitly implementing a clock using neural activity, they indirectly measure time through transitions in behavioral space. Fascinatingly, in a recent study where rats were trained to time out a particular interval of time, the authors demonstrated that the rats solved the task by developing highly stereotyped movements that spanned the target interval, suggesting a possible link to behavioral theories of timing [17]. The stigmergic strategy we observed here with our agents is therefore similar in nature to the strategy rats naturally adopted in that study. This observation suggests that animals and artificial agents may converge on similar solutions for interval timing, and this may be a consequence of shared computational constraints across both systems.

## 3.5 Exploring architectural variants

Given the difference between the behaviors of the recurrent and feedforward agents, we explored and compared the performance of some alternative architectural variants (Fig.7). The goal of this analysis was to briefly explore the sensitivity of the agent's performance to its specific setup, and future work should extend these analyses to deeper characterization of a broader range of architectures.

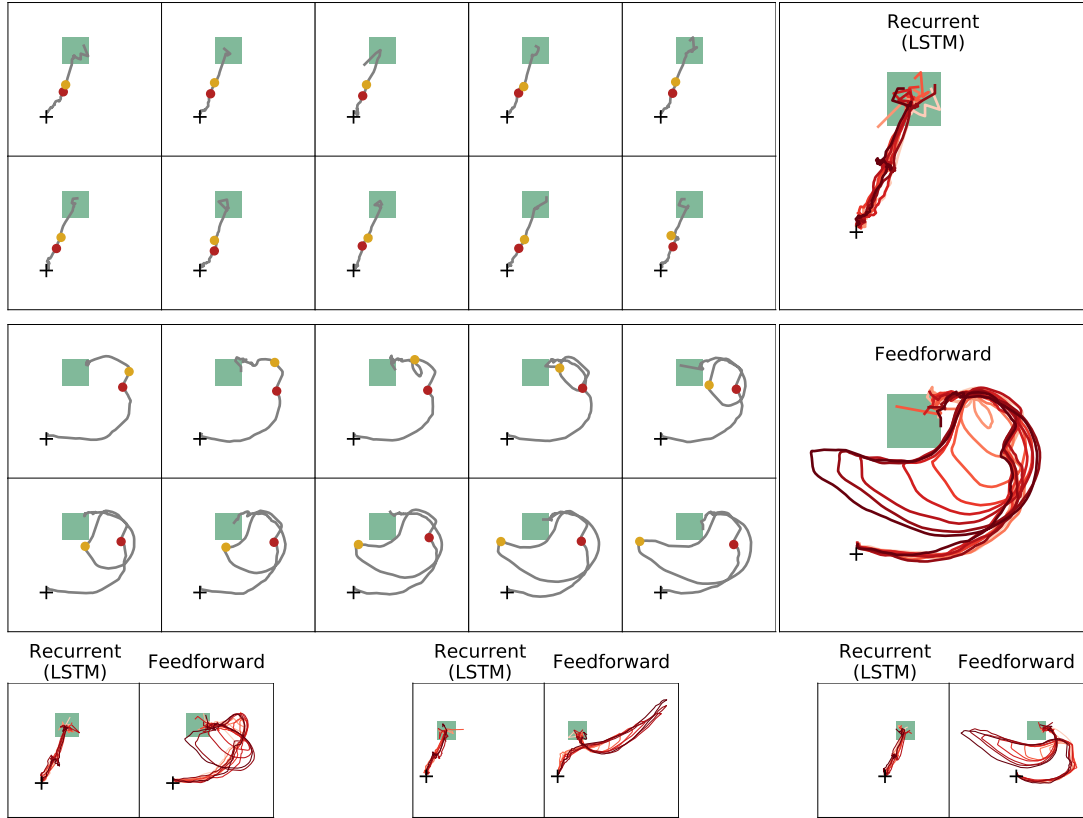

Figure 6: Agent gaze trajectories. (a) The gaze position of the agent was recorded at each time point throughout the trial for rewarded trials of varying sample interval durations. Each panel shows a schematic of the environment, with the black cross representing the central trial initiation gaze target, and the green square representing the Go target. Each subpanel shows the trajectory of gaze position over time (gray line) averaged across trials of one specific sample interval duration. For reference, the red dot corresponds to the moment when the "Ready" cue appeared, and the yellow dot to when the "Set" cue appeared. The upper left panel shows the mean over trials with the shortest sample interval, increasing rightward, with the bottom right showing the longest. The large right-side panel shows the trajectories overlaid, colored according to sample interval duration (the darkest red corresponds to the longest sample interval, i.e. the trajectory in the bottom-right small panel). (b) The same as shown in a, but for the feedforward agent. (c) Three more pairs of examples from different seeds comparing the recurrent and feedforward agents.

We first varied the number of hidden units (set at 128 throughout the study) to determine whether fewer parameters in the LSTM agent might degrade performance, or whether more parameters in the feedforward agent may augment performance. These alterations had minimal effect on the overall performance of the agents. We next trained agents in which the controller was not an LSTM or feedforward network but rather a vanilla RNN, GRU [4] or RMC (relational memory core) [26] instead. Agents with these controllers all learned the task, though the vanilla RNN and RMC exhibited some biases in performance. We proceeded to ask about the performance of a frozen LSTM: that is, its parameters were non-trainable; they were initially randomized and not changed thereafter throughout training. Interestingly, this agent learned to the same degree as the basic LSTM agent. This finding aligns with other reports that learning can occur in networks with random weights [20]. Given this apparent robustness, we then asked whether it would learn when the reinforcement learning algorithm was limited to policy and baseline updates on smaller segments of agent-environment interactions (i.e. fewer steps). In particular, we modified the agent such that the 100 steps used in backpropagation through time were divided into 10 chunks (of 10 steps each) for the sake of computing the policy gradient and baseline losses for the reinforcement learning algorithm (See A.1.1 for details). In this truncated "10-step RL," the reinforcement learning algorithm trained on episodes far shorter

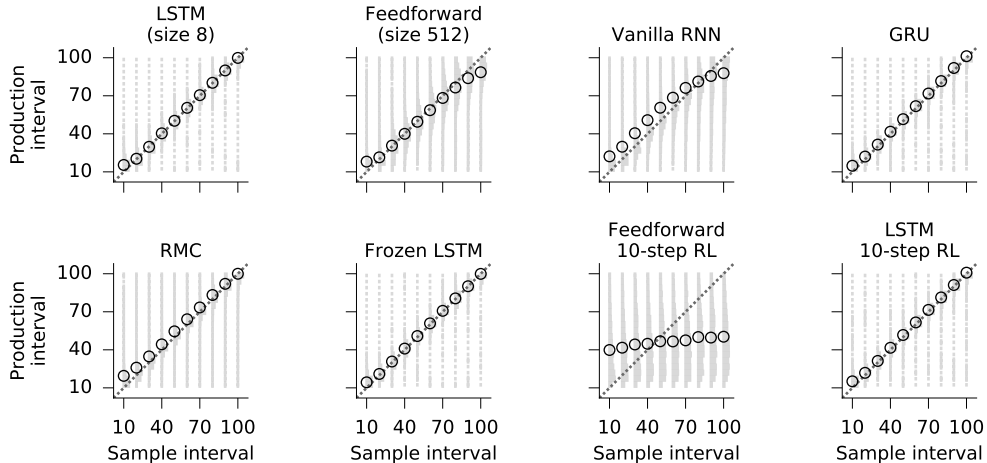

Figure 7: Performance with architectural variants. Display conventions are as in Fig. 3b. LSTM/Feedforward size indicates the number of hidden units in the controller (as compared to 128 in all prior figures). Vanilla RNN, GRU, and RMC were substituted in as replacements for the LSTM/feedforward controllers. Frozen LSTM is an LSTM controller where parameters are not trainable. 10-step RL refers to an agent that uses chunks of 10 agent-environment steps to compute policy- and baseline-gradient updates (as compared to 100 steps in all prior figures).

than the temporal intervals to be learned. We found that while the feedforward agent (which lacks backpropagation through time) was severely impaired by this alteration, the recurrent agent was not.

## 4  Discussion

Here we adapted an interval timing task from the neuroscience literature and used it to study deep reinforcement learning agents. We found that both recurrent and feedforward agents could solve the task in an end-to-end manner. We furthermore characterized differences in the behaviors of the agents at the levels of timing precision and generalization, hidden unit activations, and trajectories through action space. Recurrent agents implemented timers that could be characterized as counters in the LSTM hidden units, whereas feedforward agents developed stigmergy-like strategies that bear resemblance to psychophysical results from timing experiments in animals.

The application of neural network models to questions from experimental neuroscience can aid in our understanding of neural coding in the brain [22]. The importance of understanding interval timing in deep reinforcement learning agents has been previously recognized [25], and other work has been performed using neural networks to study time perception. For example, [16] and [27] explored computational models of time perception and its relation to environmental stimuli. In addition to the temporal reproduction task we studied here, there exist other timing tasks that are commonly used in the animal literature, such as temporal production [17] and temporal discrimination [23] tasks. Therefore, we have also generated tasks like this in PsychLab for future study, and we are open-sourcing all these tasks as part of this contribution [1].

Future work should explore the ways in which different environmental and agent architectural constraints alter the solutions of the agent. Furthermore, it will be useful to determine how findings from interval timing tasks like these, performed in controlled psychology-like environments, generalize to more complex domains where interval timing is necessary but is not the primary goal. Finally, characterizing agents' solution space for fundamental abilities like timing will be useful in designing future challenges and solutions to more complex tasks for AI. Perhaps the stigmergic behavior we uncovered in this study indicates the broader importance of deeply characterizing – and possibly controlling – agent behaviors in conjunction with their architectures when designing and studying intelligent abilities.

**Acknowledgements** We thank Neil Rabinowitz for helpful discussions on data interpretation and experiment design.

## Footnotes

[1]Available at `https://github.com/deepmind/lab/tree/master/game_scripts/levels/contributed/psychlab`

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
