[Reviews · NeurIPS 2019]

Reviewer 1



After reading the Author Feedback: The authors addressed and responded to all my concerns in an extensive manner. This is an interesting well-thought contribution, and I am happy to increase my score. Summary: In this paper, the authors investigate how deep reinforcement learning agents with distinct architectures (mainly, feed-forward vs. recurrent) learn to solve an interval timing task analogous to a time reproduction task widely used in the human timing literature, implemented in a virtual psychophysics lab (PsychLab/DeepMind lab). Briefly, in each trial the agent has to measure the time interval between a "ready" and "set" cue, and wait for the same duration before responding by moving their virtual gaze inside a "go" target; with the goal that the duration between the "set" cue and the "go" response should match the interval between "ready" and "set". Time intervals during training are drawn from a discrete uniform distribution. Perhaps not surprisingly, recurrent LSTM networks can learn and perform the task well, and their performance generalizes to unseen time intervals, both via interpolation and extrapolation. Interestingly, even feed-forward networks, which lack an explicit memory, are able to perform the task, although with degraded performance. The authors analyze the networks to understand their learnt solutions, finding the representation of a timer in the activation of LSTM units. On the other hand, the feedforward network encodes time in the gaze trajectory itself, a form of "auto-stygmergy" (that is encoding information in the environment, in this case the agent's gaze location). Originality: Medium. The investigation of internal representations of (deep) neural networks is becoming more and more common in computational neuroscience, and as far as I can tell there is nothing specifically new in the methodology here; also, previous studies have used neural networks (more or less biologically-inspired) to understand the development of representations of time in the absence of an explicit clock (notably, Karmarkar & Buonomano, 2007; a seminal reference which is missing here). Quality: High. The experiments and analysis are well-thought and sound, and the authors explore the robustness of their findings with several architectural variants. Clarity: The paper is well-structured; the text is well-written and very clear, and the images provide useful information. Significance: This is an interesting contribution potentially both for the human timing field and for machine intelligence (in that reinforcement learning agents, for most tasks of interest in the world, have to use some representation of timing; so it may be useful to understand how that develops). Major comments: This is a solid, interesting paper and a pleasure to read. I congratulate the authors for making their experimental setup available. For the purpose of reproducibility, I encourage them to also release the code used to run the analysis (whatever they can; I understand that they have limitations as noted in the reproducibility checklist), and possibly the trained networks (at least the two main ones used in the paper; not necessarily all the variants). My first comment is that the current setup does not enforce the subject to hold fixation (as most if not all psychophysical tasks involving eye movements would require the subject to do). Clearly, the neural network agents (at least the feed-forward one) are exploiting this freedom to perform the task. As a simple control experiment, the agents should be required to keep fixation within a small box surrounding the fixation cross, until the "set" cue goes off. I imagine that the prediction is that the feed-forward agents would become (almost completely) unable to perform the task; while the recurrent agent should be unaffected. More importantly, one of the results of the paper is that for the feedforward agent the standard deviation of the production interval scales linearly with the sample interval ("scalar variability"), which is taken as a signature of "biological" behavior, known in many fields as Weber's law (the main alternative would be a square-root scaling, which is a signature of counting); see lines 131-136 and Figure 4. First, the claim of linear scaling itself is somewhat dubious - as far as I understand the main evidence is presented through Figure 4a, but no statistical analysis is provided. At the very least, I suggest that the authors fit a generalized power law (e.g., a + b x^c) to the data, get a posterior over the parameters, and check that the coverage of the posterior over c is only around 1 (while I suspect that the data could be also fit well by a lower exponent). The authors also fit a model with scalar variability to the network-produced data, but a qualitatively good fit (no quantitative metric or comparison is provided) is hardly strong proof due to the well-known lack of identifiability of these kinds of models (Acerbi et al., 2014). Second, the reward function used by the authors explicitly includes a scalar term (that is, the correctness window is proportional to the sample interval, with a beta coefficient, see line 75), so it is unclear whether the scalar law potentially seen in the data (if it is there in the first place) emerges naturally, or is simply a byproduct of the chosen reward structure. The authors might test this by changing the reward function such that it uses a fixed, interval-independent window. Minor comments: line 78: The authors set beta to "8 frames", but beta is a coefficient that multiplies the time interval t_s, so I am a bit confused about the dimensionality; beta should be an adimensional scalar. Discussion: A couple of missing citations that may be relevant as related work. First, Karmarkar and Buonomano (2007) explore how neural networks can give rise to a sense of timing without an explicit clock. Also, a recent paper investigated how different task representations emerge in recurrent neural networks during short-term memory tasks, which might be worth mentioning (Orhan and Ma, 2019). Typo: line 94: followed max --> followed by max References: Acerbi, L., Ma, W. J., & Vijayakumar, S. (2014). A framework for testing identifiability of Bayesian models of perception. In Advances in neural information processing systems (pp. 1026-1034). Karmarkar, U. R., & Buonomano, D. V. (2007). Timing in the absence of clocks: encoding time in neural network states. Neuron, 53(3), 427-438. Orhan, A. E., & Ma, W. J. (2019). A diverse range of factors affect the nature of neural representations underlying short-term memory. Nature neuroscience, 22(2), 275.

Reviewer 2



Quality: I found the basic idea and result interesting, but also found the understanding of this behavior to be somewhat lacking. For example, while the timing of the agent is well fit by a Bayesian model, it is not clear why the agent would arrive at this strategy and whether it arrives at this strategy for the same reasons as biological systems. Clarity: I found this paper fairly clear and easy to understand. The basic results are clearly presented, and the authors are making their code available online. Originality and Significance: I would classify this work as original, although I’m undecided as to its significance. While it is fairly intuitive that less powerful architectures would display biases away from the optimum strategy, it is interesting to see how these biases agree with those found in biological systems. I do, however, have questions about the underlying mechanisms for these biases emerging. For example, does there not exist a set of network weights that works for the simple strategy, or does the strategy emerge and become frozen from the learning dynamics. If so, how is this strategy learned? Miscellaneous:The PDF file seems to be broken. My computer crashed several times due to looking at either page 3 or page 4 of the PDF (Perhaps figure 3), and I was not able to print these pages either. I don’t know why this is.

Reviewer 3



This is an impressive amount of work, and an interesting example of how artificial neural networks are now being studied with ideas analogous to how biological networks are treated. As with such networks though it is important to choose the right experiments to make results generalisable (see below). The paper is well written and an enjoyable read. I would have enjoyed a little more discussion of how general the results are, are they just specific to this network architecture? What has been learned about recurrent networks? Does this tell us anything about the biological brain? Training of machine very different to animal learning surely, limiting how much can be learned about animal intelligence? This is not mean a criticism, just an observation for discussion. ’Stigmergy’ solution is very specific to the details of the task, i.e. changing the location of the fixation target completely invalidates the strategy. Is the LSTM trained to be generalised across locations? Please discuss. Minor: References 19 and 20 look identical Incomplete references

[Author Response · NeurIPS 2019]

We thank the reviewers for their constructive and thoughtful reviews of our work. We are excited about these results and we are pleased that they share our sentiments.

Reviewer #1 made an excellent point regarding the fixation requirement; we considered this problem at length and have two general responses. First, we ran additional versions of our experiments in which fixation was required, and indeed all agents failed to learn this variant (which was expected given that they learn through random exploration of the action space, which is unlikely to yield fixation strategies from which to discover rewards). We also ran variants wherein the agent *trained* normally but was tested with fixation requirements. Still, the agents failed to maintain fixation at test time. From this we conclude that a carefully designed training curriculum is necessary to enable RL-based discovery of this strategy, not unlike the extended behavioral shaping procedures in the animal literature. We think this is an important next step to consider but did not implement it at this stage. More generally, we hypothesize that both agents and animals will seize upon the action spaces available to them; such as limb movements in the rodent study we cited, gaze trajectories in our study, and possibly non-experimentally monitored movements (e.g. finger tapping) in studies requiring fixation. We have included the aforementioned variants in the code we will open-source before the conference.

Reviewer #1 also made an important point regarding scalar variability. In response, we fitted a generalized power law to the data, and the coverage of the posterior over c is indeed around 1, though the best-fit value is 0.7, as the reviewer suspected. At this point we view the finding as an approximation of scalar variability with the possibility of a more complex relation to be explored. Regarding the possibility of this scaling emerging from the reward structure: this is an important point, and one we considered when setting task parameters prior to analysis; in fact, both this and the subsequent point concerning the dimensionality of beta are explained by a typo on our part: alpha and beta were swapped in the task description. In fact, alpha was set to 8 frames, and beta was set to 0, thus eliminating reward scaling (though that option exists in the task architecture in case future users choose to set it to a non-zero value). In summary, the approximate scalar variability we observe is indeed real and not related to task reward structure.

Finally, we are also releasing all the code we are able to in order to support reproducibility. The agent can be reproduced using the open-source IMPALA implementation, and we are happy to provide advice by email to anyone seeking to do so. We also thank the reviewer for pointing us to these important references that we will now discuss in the paper: the Karmarkar and Buonomano (2007) result presents important evidence that timing can be achieved without the often-proposed centralized clock mechanism; our result similarly points to strategies for timing requiring no explicit clock. The Orhan and Ma paper is also highly relevant, since our task can be framed as a particular form of short-term memory, and they address the timely topic of persistent neural activity vs. sequences in short-term memory.

Reviewer #2 raised the important consideration of learning dynamics and how the agent converges on its strategy. We always trained the agent end-to-end (from pixels to actions) and it therefore acquired its strategy through RL-based exploration of the action space and environment. Most of our work focused on placing strict *environmental* (i.e. task) constraints on the agent. We agree that an important extension of the work is to place specific constraints on the agent architecture in order to more thoroughly identify the mechanisms by which the agent develops strategies. We have attempted preliminary experiments in which we perturb the forget gates of the LSTM model causing it to be forgetful, hypothesizing that it may then converge on a different behavioral strategy. We also point to our result in which a frozen LSTM can learn the task; this demonstrated the non-necessity of trained LSTM gates to achieve the task. There is much more exploration that can be done here, and we take that to be one of the main results of our work: agents can find many solutions using the abilities given to them, so careful task design is crucial to understanding agent behavior.

In response to Reviewer #3: we have made some attempts to determine how general these results are. Figure 7 shows a sweep across various agent architectures, and so far the purely feedforward agent was the only one to strongly develop the stigmergic strategy. From this we conclude that recurrence in any form is likely to be used for timing purposes when available to the agent; whereas in its absence, agents learn more behaviorally linked strategies. In the context of biology, we agree that it is difficult to compare animals and agents. Nevertheless, we think that the similarities to some biological data may suggest that animals rely on neural strategies with relatively low memory capacity (feedforward systems being one such example). More importantly, we consider it an important cautionary tale for the study of animals: modern systems neuroscience often approaches these questions with the prior that neural activity will most directly explain cognitive phenomena, whereas results like these demonstrate that a behavioral phenotype may be a more direct mechanistic explanation that should be analyzed in concert with neural activity.

Regarding the changing of target locations: we indeed attempted these experiments, and agents failed to learn task variants with random target placements. We suspect that development of more detailed curricula will overcome this barrier. Interestingly, in a binary temporal discrimination task which we are also open-sourcing ("report whether the stimulus was short or long"), agents learned to match intervals to randomly placed colored targets, but failed when the rule was a pro-anti rule instead. We think these task complexity barriers to learning are fascinating, and one lesson from this study is the importance of detailed analyses concerning which requirements cause agent learning to break down.

[Meta-Review · NeurIPS 2019]

This paper presents an open source interval reproduction task for RL agents that is based on psychophysics tasks originally developed in neuroscience. This task is used in the paper to better understand how different RL agents solve timing tasks, for example by examining action trajectories and unit activations. This work establishes an open tool and a framework that could be used by future studies to understand computations related to timing in various RL agents. The reviewers all agreed that this paper provides a worthwhile contribution to both the machine learning and neuroscience communities. They had some initial concerns related to generality and scalar variability. But, the reviewers were happy with the author responses on the issues raised and all agreed that this was a good paper that passed the bar for acceptance at NeurIPS.